# Oral Health of COVID-19 Patients Using Extracorporeal Membrane Oxygenation: A Clinical Study of 19 Cases

**DOI:** 10.3390/jcm11010042

**Published:** 2021-12-23

**Authors:** Aya Yoshino, Yoshihiko Nakamura, Yuhei Irie, Taisuke Kitamura, Tohru Takata, Hiroyasu Ishikura, Seiji Kondo

**Affiliations:** 1Department of Oral and Maxillofacial Surgery, Faculty of Medicine, Fukuoka University, 7-45-1 Nanakuma, Jonan-ku, Fukuoka 814-0180, Japan; kondo@fukuoka-u.ac.jp; 2Department of Emergency and Critical Care Medicine, Faculty of Medicine, Fukuoka University, 7-45-1 Nanakuma, Jonan-ku, Fukuoka 814-0180, Japan; pdmxy827@yahoo.co.jp (Y.N.); iriey@fukuoka-u.ac.jp (Y.I.); taisukekitamura@gmail.com (T.K.); ishikurah@fukuoka-u.ac.jp (H.I.); 3Department of Oncology, Hematology and Infectious Disease, Fukuoka University Hospital, 7-45-1 Nanakuma, Jonan-ku, Fukuoka 814-0180, Japan; takattol@cis.fukuoka-u.ac.jp

**Keywords:** COVID-19, oral condition, ECMO

## Abstract

The oral health of coronavirus disease 2019 (COVID-19) patients in the intensive care unit (ICU) is an important issue in treatment of respiratory failure. We retrospectively investigated the oral health history of severe COVID-19 patients who received extracorporeal membrane oxygenation (ECMO) from April 2020 to December 2020 using the oral assessment guide from Fukuoka University (OAG-F). Nineteen consecutive patients (median age: 62 years) were divided into two groups according to survival (survivors, *n* = 12; non-survivors, *n* = 7). A univariate analysis revealed no significant differences between the groups in sex, age, body mass index (BMI), or the number of remaining teeth, whereas the ECMO assistance of non-survivors (median: 34 days) was prolonged in comparison to survivors (median: 8 days; *p* < 0.05). Among the factors of OAG-F, significant differences were observed between the groups in the conditions of the saliva, mucous membrane, and gingiva. The total scores in non-survivors (median: 19) were significantly higher in comparison to survivors (Median: 15.5), suggesting that the frequency of oral health deterioration was higher in non-survivors (*p <* 0.05). Taken together, these findings suggest that poor oral health is associated with mortality in COVID-19 patients receiving ECMO in the ICU.

## 1. Introduction

Coronavirus disease 2019 (COVID-19), which is caused by a novel severe acute respiratory syndrome coronavirus 2 (SARS-CoV-2), has caused respiratory failure worldwide and represents an extremely serious health crisis. While most COVID-19 patients have mild symptoms, up to one-quarter of hospitalized patients require intensive care unit (ICU) admission. These severe cases exhibit respiratory failure with an excessive systemic inflammatory reaction and multiple-organ dysfunction [1,2,3] and require oxygen supplementation and mechanical ventilation. From this perspective, the WHO interim guidelines [4] advocated venovenous extracorporeal membrane oxygenation (ECMO) to restore respiratory function in severe COVID-19 patients. Bleeding and thrombosis are serious complications associated with ECMO treatment, and oral bleeding is one of the adverse events that require special attention [5], besides prolonged mechanical ventilation, including lying in the prone-positioning, heavy sedation, and the administration of muscle blockers for several weeks. As a result, such patients are at high risk for developing bacterial ventilator-associated pneumonia (VAP). The oral management of such ICU patients is extremely important because oral care can help to reduce the incidence of VAP [6].

COVID-19 patients often experience oral health problems, such as dry mouth, mucosal blistering, mouth rash, lip necrosis, and loss of taste and smell [1,3,7] because the angiotensin-converting enzyme-2 (ACE2) and transmembrane protease serine 2 (TMPRSS2) virus receptors of SARS-CoV-2 are expressed in the oral and oropharynx tissues [8]. Poor oral health, such as periodontitis, may be an important risk factor for severe complications of COVID-19 infection in patients managed in the ICU [9,10]. During prolonged endotracheal intubation, dissemination of oral bacteria into the respiratory tract disturbs the preferable composition of the microbial community of the lung, leading to impaired systemic immunity and homeostasis [11]. Thus, oral problems can be life-threatening if left untreated.

This study aimed to evaluate the prevalence of oral complications in severe COVID-19 patients under ECMO management in the ICU and to reveal the association between oral conditions and COVID-19 severity and mortality in patients managed in the ICU.

## 2. Materials and Methods

### 2.1. Study Design

This retrospective study included 19 patients with COVID-19 who received ECMO in Fukuoka University Hospital ECMO Center between April 2020 and December 2020. In the center, all patients had received ECMO treatment and an oral assessment each day by nurses. Our hospital uses an original oral assessment guide (Oral Assessment Guide Fukuoka University Hospital: OAG-F). The OAG-F is a modified version of the oral assessment guide developed by Eilers et al. in 1988 [12] for our hospital. The palpation in OAG-F is examined with the tip of the finger instead of the tongue blade to simplify the assessment. The OAG-F examines eight items: Voice, swallow, lips, tongue, saliva, mucous membranes, gingiva, teeth, and dentures. The variables and ratings are shown in Table 1. The following data were also collected: The treatment outcome at the time of hospital discharge, sex, age, body mass index (BMI, kg/m^2^), duration of ECMO treatment, and the number of teeth.

### 2.2. Oral Health Care

Oral health care was given to all of the patients 3–8 times a day using sponge brushes, extra-soft toothbrushes, physiological saline, and no topical antiseptics. Oral moisturizers and petrolatum were also applied locally to the xerostomia. A timetable of the oral assessment and oral healthcare is shown in Table 2.

### 2.3. Statistical Methods

Statistical significance was assessed using Fisher’s exact test, the Mann–Whitney *U*-test, and the Wilcoxon signed-rank test. *p* values of <0.05 were considered statistically significant.

To investigate the relationship between the ultimate outcome in the center (life or death) or duration of ECMO and OAG-F score, we performed a multiple regression analysis that included the delta OAG-F score as an objective variable and the center mortality and the duration of ECMO as explanatory variables. The delta OAG-F score was calculated as follows: Delta OAG-F score = OAG-F score on discharge-on admission.

### 2.4. Ethical Considerations

The study was approved by the Clinical Research and Ethics Centre of Fukuoka University (No. U21-03-006).

## 3. Results

### 3.1. Characteristics of COVID-19 Patients Who Received ECMO

A total of 19 patients were included in the analysis. Table 3 displays the frequency of the selected characteristics of the study subjects. The median age was 62 years (interquartile range [IQR] 51–69 years), 84.2% of the patients were male. Twelve of the patients survived, seven died. The age, BMI, and the number of teeth of survivors and non-survivors did not differ to a statistically significant extent (*p* > 0.05). The duration of ECMO treatment was significantly longer in non-survivors.

### 3.2. Oral Conditions of COVID-19 Patients Who Received ECMO

Table 4 shows the median OAG-F scores on admission to the ECMO center. The scores of the eight items and the total score did not differ to a statistically significant extent between survivors and non-survivors (*p* > 0.05). On the other hand, the OAG-F score measured at the time of discharge from the ECMO center revealed that the saliva, mucous membranes, gingiva, and total scores were significantly higher in non-survivors (Table 5).

Figure 1 shows the change in the total OAG-F scores between admission and discharge. The total OAG-F scores of the survivors indicated a significant downward trend, whereas the scores demonstrated an increasing trend in non-survivors (*p* < 0.05). The time-course changes of the total OAG-F score were investigated (Figure 2), which demonstrated the same trend as Figure 1, although the number of samples was insufficient to analyze the data on Day 21 because the hospitalization period of some patients was shorter than 21 days.

The results of the multiple regression analysis are shown in Table 6. The ECMO center outcome was an independent factor for the elevation of the OAG-F score, the duration of ECMO was not directly related.

## 4. Discussion

This is the first preliminary report describing the oral conditions of severe COVID-19 patients who received ECMO in an ICU. According to oral assessment via the OAG-F, hyposalivation, ulceration of the mucous membranes, and gingivitis were more frequently observed in non-survivors, although there are certain limitations associated with our study, including the small sample size, the fact that the genetic characteristics of the host, and virus variants were not considered, we were not able to take into account severe cases without ECMO as a control group in our center, public standards assessment of periodontal conditions was not used due to restriction of entry to the ICU room by oral healthcare professionals, and there was some variability in the treatment regimens of the two groups. It seems that the impairment of their oral health was not due to direct infection, rather it was due to secondary manifestations resulting from life-saving invasive treatment, including orotracheal intubation, external ventilation and blood oxygenation, and pharmacotherapy with immunomodulation, as well as a lack of oral care in the ICU [1]. Of note, such oral health deterioration triggers dysbiosis of the oral microbiota, like periodontal disease, as well as systemic inflammation. These situations are often observed in patients with severe COVID-19 infection and contribute to the progression of the disease through the elevation of inflammatory cytokine levels: So-called cytokine storm syndrome [1,2,3]. In severely ill hospitalized patients, periodontitis can be a more important risk factor for complications of COVID-19—especially in patients staying in the ICU—in comparison to patients hospitalized in the general ward [9,10].

Until now, early evidence has suggested that the role of oral bacteria in the infectious pathogenesis of the respiratory tract can be explained via the aspiration of oral bacteria into respiratory organs, like microaspiration [13], which leads to susceptibility to systemic inflammation as the inflammatory cytokine levels increase from bacteremia [14,15]. Good oral hygiene is therefore essential for preventing the dissemination of oral bacteria to respiratory organs. In fact, oral hygiene interventions improve severe complications in patients with pneumonia [16], and reduce the incidence of ventilator-associated pneumonia in ICU patients [17]. With regard to COVID-19 infection, pathogenic oral bacteria infect the respiratory tract through the same mechanism. Furthermore, a novel conceptual disease model, the oral-lung axis, has been receiving increased attention [18]. That is, oral bacteria translocate to the respiratory tract, disturbing the preferable composition of the lung microbial community, leading to impaired systemic immunity and homeostasis, and an increased incidence of pneumonia [11,19]. To support this, our findings also suggested the involvement of a suppressed systemic immune response and/or susceptibility of the oral mucosa (Table 4 and Table 5).

In the field of oral medicine, the potential to manage COVID-19 complications through routine oral hygiene intervention is an attractive possibility. Many ECMO patients have impaired oral health due to their inability to perform adequate oral care because of advanced medical treatment, including orotracheal intubation, tracheostomy, and external ventilation when the treatment period is prolonged. Although oral care is basically a simple and non-invasive interventional technique for mitigating the colonization of oral bacteria, it has been shown to be more effective than using antibiotics. Oral healthcare providers should use full personal protective equipment (PPE: gloves, gowns, face shields, and surgical masks) for the management of COVID-19 patients [20], which is associated with high medical costs and waste. However, it is plausible that this approach will be indispensable for the introduction of the multidisciplinary management of COVID-19 patients who receive ECMO in the ICU [9]. Further investigations should be undertaken to investigate the role of oral healthcare in the context of associated COVID-19 complications.

## Figures and Tables

**Figure 1 jcm-11-00042-f001:**
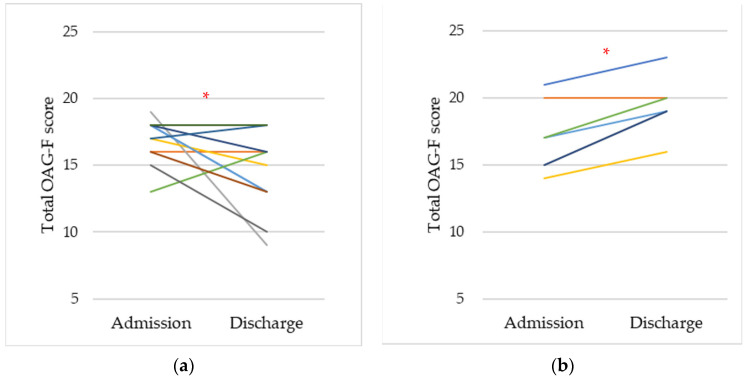
The change in the total OAG-F score between admission and discharge. (**a**) Survivors. (**b**) Non-survivors. * Asterisk indicates a statistically significant difference (*p* < 0.05).

**Figure 2 jcm-11-00042-f002:**
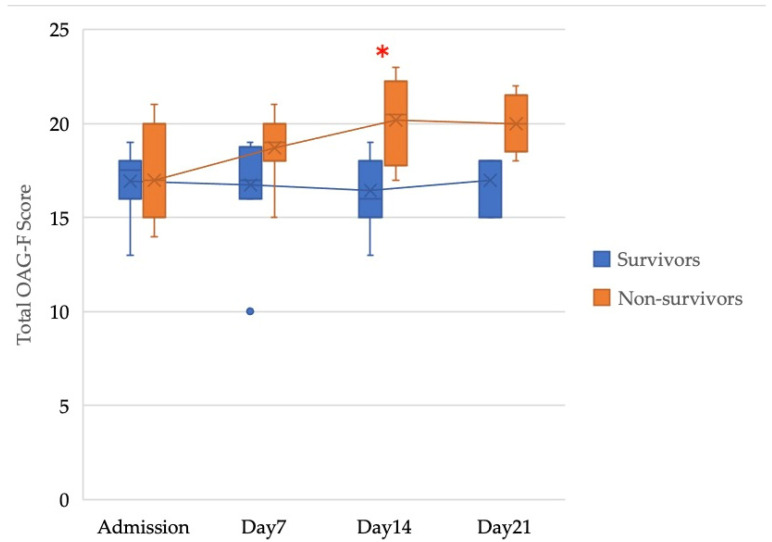
The time-course changes in the total OAG-F score. (Admission) Survivors: *n* = 12, Non-survivors: *n* = 7. (Day 7) Survivors: *n* = 12, Non-survivors: *n* = 7. (Day 14) Survivors: *n* = 7, Non-survivors: *n* = 6. (Day 21) Survivors: *n*= 3, Non-survivors: *n* = 5. * Asterisk indicates a statistically significant difference (*p* < 0.05).

**Table 1 jcm-11-00042-t001:** Scores of the eight items in the OAG-F.

Category	Tools for Assessment	Method of Measurement	Score		
			1	2	3
Voice	Auditory assessment	Converse with patient	Normal	Deeper or Rusty	Difficulty talking or painful
Swallow	Observation	Ask patient to swallow	Normal swallow	Some pain on swallow	Unable swallow
Lips	Visual/ palpatory	Observe and feel tissue	Smooth and pink moist	Dry or cracked	Ulcerated or bleeding
Tongue	Visual/ palpatory	Feel and observe appearance of tissue	Pink and moist and papillae present	Coated or loss of papillae with shiny appearance with or without redness	Blistered or cracked
Saliva	Visual/ palpatory	Observe and feel saliva	Watery	Thick or ropy	Absent
Mucous Membranes	Visual assessment	Observe appearance of tissue	Pink and moist	Reddened or coated (increased whiteness) without ulcerations	Ulcerations with or without bleeding
Gingiva	Visual/ palpatory	Gently press tissue	Pink and stippled and film	Edematous with or without redness	Spontaneous bleeding or bleeding with pressure
Teeth, Dentures	Visual assessment	Observe appearance of teeth or denture bearing area	Clean and no debris	Plaque or debris in localized areas (between teeth if present)	Plaque or debris generalized along gum line or denture bearing area

**Table 2 jcm-11-00042-t002:** Timetable of the oral assessments and oral healthcare.

Mechanical Ventilation	Total OAG-F Score	Time 0:00	1:00	2:00	3:00	4:00	5:00	6:00	7:00	8:00	9:00	10:00	11:00	12:00	13:00	14:00	15:00	16:00	17:00	18:00	19:00	20:00	21:00	22:00	23:00
No	–8										○	▲			○						○				
	9–12			□				○□		□		▲	○□			□			○□					□	
	13–			□				○□		□		▲	○□			□			○□			□			□
Yes	–12		□				○□					▲	○□						○□				□		
	13–14 or MM * > 1		□				○□			□		▲	○□			□			○□				□		
	15– or MM * = 3			□			○□			□		▲	○□			□			○□			□			□

* MM: OAG-F Score of mucous membrane; ▲: Assessment (OAG-F Scoring); ○: Brushing of teeth; □: Mucosal care.

**Table 3 jcm-11-00042-t003:** Patient characteristics.

		All Cases N = 19	Survivors N = 12	Non-Survivors N = 7
Sex ^1^	Male	16 (84.2%)	10 (83.3%)	6 (85.7%)
	Female	3 (15.8%)	2 (16.7%)	1 (14.3%)
Age ^2^		62 (51–69)	59 (39–64)	69 (59–71)
BMI ^1^	Adequate weight (<25)	8 (42.1%)	5 (41.7%)	3 (42.9%)
	Overweight/Obese (≥25)	11 (57.9%)	7 (58.3%)	4 (57.1%)
Number of teeth ^2^		27 (24–28)	27 (26–27)	28 (23–28)
Duration of ECMO treatment (days) ^2^		10 (7–29)	8 (5–11.5)	34 (12–74) *

^1^ Value represents the number (%); ^2^ Value represents the median (IQR); * Asterisk indicates a statistically significant difference (*p* < 0.05).

**Table 4 jcm-11-00042-t004:** Median OAG-F scores on admission.

	Survivors	Non-Survivors
Voice	3 (3–3)	3 (3–3)
Swallow	3 (3–3)	3 (3–3)
Lips	2 (2–2)	2 (2–3)
Tongue	2 (2–2)	2 (1–2)
Saliva	2 (2–2)	2 (2–2)
Mucous membranes	2 (1–2)	1 (1–2)
Gingiva	1.5 (1–2)	2 (1–2)
Teeth and dentures	2 (2–2)	2 (1–2)
Total	17.5 (16–18)	17 (15–20)

Values represent the median (IQR).

**Table 5 jcm-11-00042-t005:** Median OAG-F scores on discharge.

	Survivors	Non-Survivors
Voice	3 (2–3)	3 (3–3)
Swallow	3 (1–3)	3 (3–3)
Lips	2 (1–2)	2 (2–3)
Tongue	2 (2–2)	2 (2–2)
Saliva	2 (1.25–2)	2 (2–3) *
Mucous membranes	1.5 (1–2)	2 (2–3) *
Gingiva	1 (1–1.75)	2 (2–3) *
Teeth and dentures	1.5 (1–2)	2 (1–2)
Total	15.5 (13–17.5)	19 (19–20) *

Values represent the median (IQR). * Asterisk indicates a statistically significant difference (*p* < 0.05).

**Table 6 jcm-11-00042-t006:** Multiple regression analysis for OAG-F score elevation.

Variable	Coefficient	95%CI *		*p* Value
		Lower	Upper	
ECMO center outcome (dead)	3.295	1.361	5.229	0.002
Duration of ECMO (days)	−0.053	−0.128	0.022	0.154

* CI: confidence interval.

## Data Availability

Not applicable.

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
