# Peer review of "Oral Health of COVID-19 Patients Using Extracorporeal Membrane Oxygenation: A Clinical Study of 19 Cases"

_jcm, 2021, doi:10.3390/jcm11010042_

Round 1

Reviewer 1 Report

The title

1.Oral Conditions of COVID-19 Patient Using Extracorporeal Membrane Oxygenation: Clinical Study on 19 Cases Could be replaced by

Oral health of COVID-19 Patient Using Extracorporeal Membrane Oxygenation: Clinical Study on 19 Cases

2.It should be mentioned that mainly the research on COVID-19 host genetics and compile genetic variants associated with susceptibility to COVID-19 and disease severity and the oral health is with secondary meaning. 3. Severity of periodontitis is not associated  with severity of pneumonia 
4. Parodontopathogens  are not mentioned and were not isolated fin pneumonia cases and 

Author Response

Thank you for your thoughtful consideration about our manuscript.

  1. The title was revised accordingly.
  2. Lines 134-135: Sentences ‘’the fact…not considered,‘’ were added.
  3. As suggested by the reviewer, there is not a one-on-one relationship between severe periodontitis and severe pneumonia. However, Marouf et al. reported that periodontitis could be a risk factor for COVID-19 complications. Our finding also suggested that poor oral health is associated with mortality in COVID-19 patients receiving ECMO.
  4. In this study, bacteriological examinations were not performed. These issues will be considered in a future study.

Reviewer 2 Report

Dear authors,
Your article presents an interesting insight into the correlation between oral pathology and SARS-CoV-2 infection.
However, these data must be supported by other subsequent studies, and naturally by expanding the number of subjects examined.
Furthermore, I believe this information may prove useful to ICU practitioners in managing SARS-CoV-2 patients.

Author Response

The authors truly appreciate the reviewer's quite positive evaluation and instructive comments on our present study.

Reviewer 3 Report

The research is well designed and carried out, and it is very actual and up-to-date.

Abstract: it is a good summary of the paper, and it it well organized.

Introduction contains enough background informations and adequate references.

Materials and methods are clearly described, but you should consider to add  a figure (or a flow chart) of your OAG-F exam, in order to have a schematic summary for readers.

Results: statistical analysis is carried out in a proper way, but the samples involved few participants. Considering your research, it is a great achievement, but I appreciate that you state it as a drawback of your work in discussion. 

I suggest to consider the following papers, in order to extend a little bit the discussion section:

1) https://www.thieme-connect.de/products/ejournals/abstract/10.1055/s-0040-1715923

2) https://www.mdpi.com/1999-4915/11/5/463

Conclusions are adequate, I hope that in the future you can continue with further investigations.

Author Response

The reviewer’s careful reading is appreciated.

  1. We agree with the reviewer and have added the summary figure of this study protocol as Figure 2. And the following sentences were added.
    Line 66 “each day”
    Lines 72-73 ”This study … the center.”
    Lines 79-80 ”A time table… in Figure 2.”
  2. Lines 134-135 : We agree with the reviewer’s suggestion and have described the limitation of our own results in the Discussion.
  3. As the reviewer commented, we have added the reference to D’Amico C et al, 2001 as Ref. #21.

Reviewer 4 Report

With "Oral conditions of COVID-19 Patient(s?) using extracorporeal membrane oxygenation: clinical study on 19 cases" the authors deal with a clinically relevant aspect with regard to the current pandemic. In spite of – as the authors duly mention in their limitations section - small numbers of this retrospective 2-arm cohort  study (19 patients overall, with 12 survivors and 7 non-survivors in the respective arms) the authors draw the conclusion that oral health deterioration is associated with non-survival in ECMO patients. The topic is of good clinical relevance, as poor oral health doubtlessly is a risk factor for inducing/aggravating complications such as pneumonia and thus enhance mortality.

The article is overall clearly written in apparently good English, and is formally well structured 

Nevertheless, the article at present shows some major  drawbacks which need to ruled out:

1st: the authors so far miss to point out convincingly which is the really new information given by the article – they state that this is the first preliminary report describing the oral conditions of severe Covid-19 patients who received ECMO – which may apply well for ECMO, but the effect of oral health on "severe cases" in essence was already described e.g. by the Slovakian working group of Hackova (including review of the literature with over 108 cases analysed),also by Marouf and co-workers from Quatar, who could show that periodontitis (and this is the essential factor, cf. 3rd) has an OR of 8.81 for lethal outcomes in COVD-19 cases  and that the patients with parodontitis show significantly higher levels of leukozytes, D-Dimers, and C-reactive protein (essential markers for the topic of oral health which are unfortunately missing in the present publication.

- The authors should therefore point out which is the essentially new information in contrast to these recent publications (except for explicitly focussing on "ECMO" instead of "severe cases" with long-term intubation, which, however – in order to allow for founded conclusions – basically would require a control group made up of severe cases without ECMO treatment

2nd:: another limitation is  a “hen and egg problem” the paper is dealing with. Whereas the Kanagawa group of Sakaguchi and co-workers had already pointed out that the oral cavity is an entrance port for SARS-CoV-2, the oral cavity problems observed in the ECMO patients of the present paper will rather be affected by the ICU circumstances with long-term reduction of oral hygiene, anticoagulation, pharmacologcical treatment (e.g steroids, cf. RECOVERY Collabortaive group) etc. – which, however, are not assessed as potentially relevant confounding parameters in the present study. More importantly, however, the odds ratio (if it would be considered) for non-survivorship would most probably also show a strong correlation to duration of ECMO viz. stay on ICU – so it is no surprise to find that those patients staying longer under ECMO will show more oral deterioration. This, however, cannot be attributed to any other factors (such as ECMO itself), then, as long as these factors have not been shown to be independent risk factors – therefore, the authors should go for some additional statistical analyses if they want to give really new information (i.e. exceeding the above quoted  papers). This even more so applies with regard to the high homogeneity of survivors and non-survivors for number of teeth, age. etc.

3rd: the OAG-F assesses “teeth”(viz. is basically plaque) and gingivitis (the procedure is traceably lined out in figure 1 but – and this is essential -  does not assess periodontitis in a specific way e.g. by measuring depths of pockets etc.: therefore, the OAG-F – which in addition was not even performed by health care professionals trained for oral assessment (i.e. dentists, OMFS)  but by the nurses in ICU. According to periodontal publication standards, a valid assessment of oral health should be based on established parameters such as SBI, PBI etc (which of course is not really feasible under COVID-19 ICU conditions, but at least needs to be addressed as a major limitation of the study design, then)

4nd: As the OAG-F was  performed apparently twice (or more often? Please specify timepoints) on admission to and end of treatment with ECMO, the authors thus basically compare apples and oranges (i.e. not taking the different durations for ECMO into account). So far, data for comparable timepoints are missing, which may lead to a completely different result as suggested by figure 2 (apparenty just comparing start and endpoint). This may be grossly misleading, then.

Typos as noticed: lines 42 40- this sentence is incomplete, so may perhaps need a colon instead of dot before besides?

Author Response

The reviewer pointed out that the nature of our problem is not totally novel in our conclusion because there are some limitations and inappropriate research designs present in this 2-arm case-control study. We have tried to address all of the points raised by the reviewer to improve the quality of the manuscript upon revision.

  1. We agree with the reviewer’s suggestion and considered severe cases without ECMO treatment as a control group. Unfortunately, however, research data were not gathered in our institution, the ECMO Center because ECMO was given to all of the patients. These issues will be considered in a future investigation.
  2. We deeply agree with the reviewer regarding the ‘’hen and egg problem’’ between oral deterioration and impairment of general condition with staying longer under ECMO in our study. Unfortunately, in the study, we had to conduct this study based on a small sample size and therefore a univariate analysis was applied to our clinical study. As a result, it was not possible to perform an additional statistical analysis including a logistic regression analysis. Instead, to take advantage of this limitation, we originally focus on the oral conditions of COVID-19 patients who received ECMO from the perspective of ‘life or death’, which is the ultimate result and the most important matter in the field of critical care, with the elimination of elements of the treatment time course such as the duration of ECMO. We found that oral deterioration was more frequently observed in non-survivors. From this point of view, we believe our clinical observational study contains novel findings despite the fact that some researchers have previously described a possible association between COVID-19 complications and the presence of periodontitis in the ICU.
  3. In Japan, COVID-19 infection was designated by government ordinance, and entry to the ICU room is restricted: only dedicated ICU staff are permitted to enter the ICU room; oral healthcare professionals are not permitted. Therefore, the valid assessment of oral conditions cannot be performed according to periodontal publication standards. For this reason, we used easy assessments, such as OAG-F, which can be applied by non-oral healthcare professionals.
  4. As the reviewer commented, there are various limitations associated with our study. To obtain a deeper understanding, we added a summary of the oral assessment and care protocol to our new Figure 2, although there was some variability in the treatment regimens of the 2 groups, including pharmacotherapy (antibiotics, steroids, etc.) and the duration of orotracheal intubation and ECMO.
  5. Line 40: The reviewer’s careful reading is appreciated. We have corrected several grammatical and spacing errors that were raised by the reviewer.

Round 2

Reviewer 4 Report

With “oral health of COVID-19 Patients using extracorporeal membrane oxygenation: clinical study of 19 cases” the authors submit a revised version, which, however, unluckily does not address the reviewer’s major criticisms as pointed out in detail in the previous  review.

Based on their findings, the authors give a detailed rebuttal to the reviewer’s criticisms but nevertheless do not alter the non-surprising conclusion that oral health deterioration is associated with non-survival in ECMO patients. The major problem in this context is that the authors did not consider the recommendation given in the previous review to consider duration of ICU stay as clinically highly relevant confounder to be expected besides severity of the illness.

As long as this critical point has not been addressed adequately (cf. requested modification of figure 3) and statistical comparison of the two groups at comparable timepoints taking duration of ICU stay into account, the paper cannot be recommended for publication

In spite of mostly comprehensible rebuttals the article, therefore, is flawed still by some major drawbacks which still need to ruled out:

Therefore, please consider the previous review:

Regarding point number 1: quotation from previous review:

1st: the authors so far miss to point out convincingly which is the really new information given by the article – they state that this is the first preliminary report describing the oral conditions of severe Covid-19 patients who received ECMO – which may apply for ECMO, but the effect of oral health on severe  was in essence already described e.g. by the Slovakian working group of Hackova (including review of the literature with over 108 cases analysed) and by Marouf and co-workers from Qatar, who could show that periodontitis (and this is the essential factor, cf. 3rd) has an OR of 8.81 for lethal outcomes in COVD-19 cases  and significantly higher levels of leukocytes, D-Dimers, and C-reactive protein (markers which are missing in the present publication
Recommendation previous review: The authors should therefore point out what are the essentially new information in contrast to these recent publications (except for explicitly focussing on ECMO instead of severe cases with long-term intubation, which, however – in order to allow for founded conclusions – basically would require a control group made up of severe cases without ECMO treatment).)

Rebuttal #1 by the authors:
We agree with the reviewer’s suggestion and considered severe cases without ECMO treatment as a control group. Unfortunately, however, research data were not gathered in our institution, the ECMO Center because ECMO was given to all of the patients. These issues will be considered in a future investigation.

Answer of the reviewer: then this drawback (lack of group for comparison) needs to be addressed under limitations of the study; in addition, the ECMO data need to be compared to the literature data given for severe cases, as suggested in the previous review.

Recommendation:
The reviewer would strongly recommend to take up this criticism and a) address these papers in the discussion and b) to address this major drawback under limitations of the study

Regarding point number 2: quotation from previous review:

2nd: another limitation is  a “hen and egg problem” the paper is dealing with. Whereas the Kanagawa group of Sakaguchi and co-workers had already pointed out that the oral cavity is an entrance port for SARS-CoV-2, the oral cavity problems observed in the ECMO patients of the present paper will rather be affected by the ICU and long-term reduction of oral hygiene, anticoagulation, pharmacological treatment (e.g. steroids, cf. RECOVERY Collaborative group) etc – which, however, are not assessed as potentially relevant confounding parameters in the present study. More importantly, however, the odds ratio (if it would be considered) for non-survivorship would most probably show a strong correlation to duration of ECMO viz. stay on ICU – so it is no surprise to find that those patients staying longer under ECMO will show more oral deterioration. This, however, cannot be attributed to any other factors (such as ECMO), then, as long as these factors have not been shown to be independent risk factors – therefore, the authors should go for additional statistical analyses if they want to give really new information (i.e. exceeding the above quoted  papers. This even more so applies with regard to the otherwise high homogeneity of survivors and non-survivors for number of teeth, age. etc.

Rebuttal #2 by the authors:

We deeply agree with the reviewer regarding the ‘’hen and egg problem’’ between oral deterioration and impairment of general condition with staying longer under ECMO in our study. Unfortunately, in the study, we had to conduct this study based on a small sample size and therefore a univariate analysis was applied to our clinical study. As a result, it was not possible to perform an additional statistical analysis including a logistic regression analysis. Instead, to take advantage of this limitation, we originally focus on the oral conditions of COVID-19 patients who received ECMO from the perspective of ‘life or death’, which is the ultimate result and the most important matter in the field of critical care, with the elimination of elements of the treatment time course such as the duration of ECMO. We found that oral deterioration was more frequently observed in non-survivors. From this point of view, we believe our clinical observational study contains novel findings despite the fact that some researchers have previously described a possible association between COVID-19 complications and the presence of periodontitis in the ICU.

Answer of the reviewer: unluckily, the reviewer cannot accept this explanation, as the authors report that the oral assessments were performed every day. So, it should be possible to do statistics comparing different timepoints, as it is obvious that the timepoints start and end of stay of ICU may be inadequate for differentiation of oral health between survivors and non-survivors: both groups need to be compared at equal timepoints: if then (!) the non-survivors show a more severe deterioration, oral health should be a predictor (which then should be shown via odds ratios).

Recommendation:
The reviewer would strongly recommend to take up this criticism and do additional evaluation od apparently available data  (daily evaluation of oral health!) and re-evaluate, then. Unless submitting these data, the article cannot be recommended for publication, because it still may compare apples and oranges. The authors need to rule out this major criticism

Regarding point number 3: quotation from previous review:

3rd: the OAG-F assesses “teeth”(viz. is basically plaque) and gingivitis (the procedure is  traceably lined out in figure  but – and this is essential -  does not assess periodontitis in a specific way e.g. by measuring depths of pockets etc.: therefore, the OAG-F – which in addition is not even performed by professionals for oral assessment (i.e. dentists, OMFS)  but by the nurses in ICU. According to periodontal standards, a valid assessment of oral health should be based on established parameters such as SBI, PBI etc (which of course is not really feasible under COVID-19 ICU conditions, but at least needs to be addressed as a major limitation of the study design)

Rebuttal #3 by the authors:

In Japan, COVID-19 infection was designated by government ordinance, and entry to the ICU room is restricted: only dedicated ICU staff are permitted to enter the ICU room; oral healthcare professionals are not permitted. Therefore, the valid assessment of oral conditions cannot be performed according to periodontal publication standards. For this reason, we used easy assessments, such as OAG-F, which can be applied by non-oral healthcare professionals.

Answer of the reviewer: the reviewer already gave some equivalent suggestion regarding  this point in the previous review.

Recommendation: The reviewer would strongly recommend to take up this criticism and address the rebuttal given in the limitations section or to critically discuss the assessment drawbacks in the discussion part 

Regarding point number 4: quotation from previous review:

4nd: As the OAG-F was  performed apparently twice (or more often? Please specify timepoints) on admission to and end of ECMO, the authors thus compare apples and oranges (i.e. not taking different durations for ECMO into account). So far, data for comparable timepoints are missing, which may lead to a completely different result as suggested by figure 2 (apparently just comparing start and endpoint. This needs to be made transparent by comparing correlating timepoints of assessment (i.e. graph showing the development over the single days in order to allow for comparison at identical timepoints, not overall “start” and “end”, only)  

Rebuttal #4 by the authors:

As the reviewer commented, there are various limitations associated with our study. To obtain a deeper understanding, we added a summary of the oral assessment and care protocol to our new Figure 2, although there was some variability in the treatment regimens of the 2 groups, including pharmacotherapy (antibiotics, steroids, etc.) and the duration of orotracheal intubation and ECMO.

Answer of the reviewer: the essential drawback is the new figure 3 – as pointed out above, the start and endpoint comparison is not adequate with regard to the conclusion drawn, as duration is a most probable predictor:

Recommendation: cf. above, please add additional data for comparable timepoints and add statistics.

Author Response

Thank you for your thoughtful consideration and recommendation about our manuscript. According to the reviewers' instructive comments, all of the criticisms have been duly addressed in this revised version. We believe it is now suitable for publication in Journal of Clinical Medicine.

1-a.  As suggested by the reviewer, we have added data and the results of a multiple regression analysis as Table 4.

The following sentences were added.
Lines 117-119 “The results of…. “

1-b.  We have added the following sentences.
Line 62 ”In the center, all patients had received ECMO treatment…”*

2. We agree with the reviewer and have added the time-course changes (Admission, Day 7, Day 14, Day 21) as Figure 4. And the following sentences were added.
Lines 113-116 “The time-course …. “

3. To obtain a deeper understanding, the time-course changes in the total OAG-F score were added in our new Figure 4, in accordance with the reviewer’s suggestion.

The following sentences were added.
Lines 164-167 “we were not able to …. “

5. As suggested by the reviewer, we have added data and the results of a multiple regression analysis as Table 4, which have also improved the quality of this work.